# Fabrication, Structure, Performance, and Application of Graphene-Based Composite Aerogel

**DOI:** 10.3390/ma15010299

**Published:** 2021-12-31

**Authors:** Dequan Wei, Xiang Liu, Shenghua Lv, Leipeng Liu, Lei Wu, Zexiong Li, Yonggang Hou

**Affiliations:** College of Bioresources Chemical and Materials Engineering, Shaanxi University of Science and Technology, Xi’an 710021, China; Weidequan@hotmail.com (D.W.); 4233@sust.edu.cn (L.L.); wuleistarry@gmail.com (L.W.); lizexiong0622@sohu.com (Z.L.); houyonggang@sohu.com (Y.H.)

**Keywords:** aerogel, applications, composite, graphene, graphene oxide, inorganic nanoparticle, natural polymer, reduced graphene oxide, structure, synthetic polymer

## Abstract

Graphene-based composite aerogel (GCA) refers to a solid porous substance formed by graphene or its derivatives, graphene oxide (GO) and reduced graphene oxide (rGO), with inorganic materials and polymers. Because GCA has super-high adsorption, separation, electrical properties, and sensitivity, it has great potential for application in super-strong adsorption and separation materials, long-life fast-charging batteries, and flexible sensing materials. GCA has become a research hotspot, and many research papers and achievements have emerged in recent years. Therefore, the fabrication, structure, performance, and application prospects of GCA are summarized and discussed in this review. Meanwhile, the existing problems and development trends of GCA are also introduced so that more will know about it and be interested in researching it.

## 1. Introduction

### 1.1. Graphene-Based Composite Aerogel

Aerogel, a highly porous material with low density and high specific surface area, is obtained by replacing the liquid in wet gel with gas without significantly changing the structure and volume of the gel network. Graphene-based composite aerogel (GCA) is composed of graphene and its derivatives graphene oxide (GO) and reduced GO (rGO) with other matrix materials. Its functions are mainly derived from graphene and its derivatives (graphene-based materials), while its structure and volume stability are mainly determined by other matrix materials [1]. The research results indicate that GCA has lower density, higher porosity, smaller pore diameter, larger specific surface area, and more stable morphology compared to general aerogels, but more importantly it has some unique characteristics, such as higher heat resistance, better electrical conductivity, and higher absorbability of metal ions [2,3]. Therefore, GCA can be used not only as a thermal insulation, sound insulation, damping, and adsorptive material, but also as an electrode material for sensors and energy storage devices [4], which has become a research hotspot and attracted people’s attention in recent years. Figure 1 shows the structure, properties, and application of GCA, and Figure 2 displays a structural schematic of graphene-based materials [5]. 

### 1.2. Preparation Principle of GCA

The preparation principle of GCA includes three key processes: sol, gel, and drying. In the sol-gel process, the reactants are uniformly mixed and reacted in the liquid phase to form clusters of different structures. The sol of graphene-based materials and matrix may be obtained by chemical vapor deposition (CVD), hydrothermal reaction, chemical reduction, and vacuum carbonization [7]. Then composite gel is constructed by the self-assembly, chemical cross-linking, template method and 3D printing. The gel contains a large amount of water or other solvents, up to more than 90%, with stable volume and no fluidity [8]. Then the solvent is removed from the gel by freeze-drying or supercritical drying to obtain GCA. The GCA always maintains higher porosity and larger specific surface area and has a similar network structure consisting of graphene-based and other matrix materials. Table 1 displays the preparation principle of GCA.

### 1.3. Current Research Situation

Figure 3 shows the number of papers published on GCA from 2010 to 2021, indicating that the research began in 2010 and the number of papers grew exponentially in the last 10 years. According to the literature reports, GCA is usually composed of graphene and its derivatives and matrix materials. The properties and functions of GCA are mainly determined by graphene and its derivatives, while the porous structure and stability are mainly determined by matrix materials. The matrix materials include inorganic nanomaterials, synthetic polymers and natural polymers. Therefore, the research process on the fabrication method, material selection, structure construction, performance and application design of GCA is very complex, and it is necessary to summarize and guide based on these research results. Currently there is also a lack of targeted summary articles. In this review, we focus primarily on reviewing the fabrication of GCA with inorganic nanomaterials, synthetic polymer, and natural polymers along with its structure, performance, and applications. 

## 2. Composition of GCA

GCA usually consists of graphene-based materials and inorganic 2D nanosheets materials, inorganic nanoparticle materials, synthetic polymers, and natural polymers. 

### 2.1. 2D Nanosheets/GCA

Inorganic 2D nanosheet materials can be compounded with graphene-based materials to prepare aerogels with better properties, including Ti_3_C_2_T_x_ (MXenes), transition metal sulfide (TMD), metal organic framework (MOF), hexagonal boron nitride (h-BN), layered double hydroxide (LDH), perovskite and black phosphorus (BP), and other hot materials [26]. 

Mxenes is a kind of flexible 2D nanosheet with light weight, high electrical conductivity, high surface activity, and excellent electrochemical properties. Li et al. [10] prepared GCA by forming MXenes@GO composites by electrospinning, and the GCA as a wave-absorbing material had the advantages of light weight and elongated attenuating paths. The preparation process and mechanism are shown in Figure 4. 2D MoS_2_ nanowires were fixed on the 2D rGO nanowires by 3D printing technology, which could solve the disadvantage of low conductivity of MoS_2_ and prepare GCA for sodium ion batteries. [9] The introduction of 2D nanomaterials in GCA has aroused widespread interest, but reasonable design steps are needed to achieve superior properties of GCA, including the need to consider the problem of easy agglomeration in the composite process.

### 2.2. Inorganic Nanoparticle Materials/GCA

Inorganic nanoparticles, such as SnO_2_, SiO_2_, TiO_2_, IO, etc. [27], have been widely used to prepare GCA. Nanoparticle GCA can be prepared by surface modification of inorganic nanoparticles or as a composite with graphene and hydrothermal assembly [28]. Cheng et al. [13] reported that TiO_2_/GCA prepared by template-free method had excellent electrochemical properties and could be used as lithium battery anodes and high-performance energy storage. In addition, the solvothermal method of depositing ZnO nanoparticles on graphene nanosheets to prepare ZnO/GCA (Figure 5) showed good thermal conductivity performance, a porous network structure, and high specific surface area, and had greatly increased contact with gas, and these characteristics make ZnO/GCA suitable for preparing gas-sensing materials [14]. IO/GCA was prepared by the in situ growth method in a high-gravity reactor, with good catalytic efficiency for catalyzing the photo-Fenton reaction [16]. Graphene, combined with the good electronic, optical, and thermal conductivity of inorganic nanoparticles for preparing aerogels, broadens its potential applications in batteries, catalysis, sensors, and so on.

### 2.3. Synthetic Polymer/GCA

The weak strength of pure graphene aerogels greatly limits their practical application. However, this situation can be changed by forming composite GCA with some synthetic polymers, such as PE, PVA, PDMS, PANI, PAM, PM. Synthetic polymer/GCA has higher strength, lower density, larger specific surface area, and good strength and electrochemical properties, and can make sensors, adsorbents, catalysts, and other items [29]. As shown in Figure 6, Xiang et al. [17] prepared PVA/GCA by using *γ*-oxo-1-pyranobutyric acid (OPBA) as adhesive to hold the PVA coating and graphene skeleton together. The GCA can be used as a pressure response sensor due to its compressibility and deformation recovery. Zhang et al. [18] produced PDMS/GCA aerogel by permeation of PDMS into the interior of graphene aerogel to obtain aerogel with ultra-high electrical conductivity (1 S·cm^−1^), thermal conductivity (0.58 W·m^−1^·K^−1^), high hydrophobicity (contact angle 135°), and excellent strength and thermal stability. An et al. [19] deposited polyaniline (PANI) into porous graphene microspheres to make conductive spherical PANI/GCA, and the inclusion of PANI enhanced the graphene network and made the microspheres more resistant to deformation (Figure 7), which had the characteristics of shrinkage after water loss, recovery after dissolution, high specific capacitance, and good cycle stability, so they can be used as porous electrodes for energy conversion.

In short, the disadvantage of a weak structure in the reapplication of 3D graphene can be solved by preparing synthetic polymer modified GCA in various shapes, such as sol-gel, immersion, hydrothermal reaction, or chemical weather deposition. With better strength, adsorption, and electrical properties, the new synthetic polymer/GCA has great development prospects. Therefore, in the future, new forms of synthetic polymer/GCA should be developed by seeking new synthetic polymers and preparation methods for better performance.

### 2.4. Natural Polymer/GCA

The natural polymers mainly include cellulose, starch, chitosan, sodium alginate, carrageenan, and pectin. Aerogels of natural polymers have abundant resource advantages, good biocompatibility and biodegradability, and can be exploited in medicine, environmental engineering, and food packaging [30]. The introduction of graphene-based materials not only improves the shortcomings of low mechanical strength, poor brittleness, and poor stability of natural polymeric aerogels, but affects their structure and properties.

#### 2.4.1. Cellulose/GCA

Cellulose is the most abundant natural polymer in nature [31]; it is divided into cellulose nanocrystal (CNC), [32] cellulose nanofiber (CNF), and bacterial cellulose (BC) according to the formation conditions and sources [33].

Cellulose/GCA can be prepared by a self-assembly process [34], and the formation mechanism is shown in Figure 8. Mi et al. [21] obtained cellulose/GCA by bi-directional freezing and CVD (Figure 8). This composite aerogel presented a porous structure with ultra-low density (0.0059 mg·cm^−3^) and high surface area (47.3 m^2^·g^−1^), and had good selective adsorption effect for oil. Using boric acid (BA) as cross-linker of GO and carboxymethyl cellulose (CMC), Ge et al. [22] synthesized CMC/GCA by the ice template method (Figure 9). The results showed that the GO content had a significant influence on the morphology and strength of the aerogel. When the content of GO reached 5 wt%, the strength was excellent, and thermal conductivity (0.0417 W·m^−1^·K^−1^) was similar to that of polystyrene foam (0.03–0.04 W·m^−1^·K^−1^). There is a good interaction between cellulose and GCA with the 3D structure, which provides a way to improve the mechanical properties of aerogels. The self-assembly process, CVD method, and template method can introduce cellulose into graphene to produce aerogels with better mechanical properties, thermal insulation, and energy storage.

#### 2.4.2. Starch/GCA

Starch is one of the most common biopolymers and usually divided into amylose and amylopectin according to the chemical structure. Starch in its natural form can easily form a gel without a cross-linking agent [35]. Chen et al. [23] developed a simple and rapid method for preparing porous starch/GCA. The aerogels can be used as electrode materials to manufacture flexible and gel-type symmetrical supercapacitors with excellent capacitance performance and high energy density (Figure 10). Starch/GCA was prepared by chemical reduction and self-assembly of nanocrystalline starch, which has higher mechanical properties, capacitance performance, and adsorption capacity.

#### 2.4.3. Chitosan/GCA

Chitosan (CS) is extracted and processed from the shells of common arthropods [36]. The aerogel prepared by CS can be exploited as an adsorbent for sewage purification, but its strength is poor. Therefore, it must be modified to increase its strength. Interestingly, the introduction of graphene-based materials can improve not only the strength and stability of the CS aerogel, but also the purification efficiency and degree. Using solution mixing and injection methods, CS/GCA of various shapes can be prepared and applied in many fields [37], such as CS/GCA microspheres and membranes (Figure 11) [38]. In addition, both graphene and CS are used as the basic skeleton or filling material of GCA, which significantly improve the aerogel’s properties. When CS was grafted onto a GO skeleton, the GCA with more ordered mesoporous and the adsorption capacity was significantly superior to that of pure CS aerogel [24]. CS/GCA has enough strength and stability and excellent absorbability for heavy metals, dyes, and organic solvents. Therefore, CS/GCA will still be a hot spot of research and application in the future. 

#### 2.4.4. Sodium Alginate/GCA

Sodium alginate (SA), an anionic polysaccharide with hydroxyl and carboxyl groups, comes from algae. It is widely used in medical stents, controlled-release drug carriers, and food thickeners, and also as a flocculant for treating wastewater. SA has different solubility affected by pH value. When the pH is less than 4, SA is insoluble, and when the pH is between 6 and 9, SA is a viscous solution. Although it is easy to form film and gel, SA-polysaccharide aerogel has weak strength and stability. Therefore, many studies have focused on functional modification and composite hybridization to overcome the limitations of the aerogel.

SA can be physically blended with graphene and chemically modified to improve its strength compared to the brittleness and easy collapse of pure SA aerogel (Figure 12) [31]. Shan et al. [25] applied in situ cross-linking to prepare SA/GCA, and the introduction of GO improved the uniformity of the spherical morphology and the efficiency and capacity of phosphorus ion adsorption in sewage, and was applied to remove phosphine pollution in water. In particular, the introduction of graphene can increase electrical conductivity and broaden the application of SA aerogel to other fields, such as its use as electrode material for supercapacitors and as a biomass aerogel catalyst with good catalytic activity.

## 3. Structure of GCA

### 3.1. Porous Structure of GCA

GCA has a porous network structure and low density. Pore structure is closely related to strength, thermal performance, and electrochemical performance. In addition, GCA can uniformly disperse the graphene nanolayers in the matrix in the form of a single layer or a few layers to reduce agglomeration. Different pore diameters of aerogels also affect their density, pore volume, and specific surface area. For example, pore volume is the volume of pores per unit mass of aerogel and is related to the inner and outer diameter. With the same inner diameter, the pore volume decreases along with the increase of the outer diameter, and with the same outer diameter, the larger the inner diameter, the larger the pore volume. Like specific surface area, pore volume is another important factor affecting the surface adsorption and loading properties of aerogels. The structural parameters, including specific surface area, density, and pore size of GCA, are listed in Table 2.

In terms of the morphology of aerogels, GCA can be roughly divided into two categories: porous structure and mesoporous core–shell structure. The common porous structures include honeycomb, microsphere, etc. The local compressibility and elasticity of aerogel honeycomb structure were well studied by density functional theory (DFT) [45]. The honeycomb structure [46] has the characteristics of ultra-low density, super-elasticity, good electrical conductivity, and high energy absorption efficiency (Figure 13), which is related to its inherent mechanical properties and elasticity [47]. By adjusting the method and process parameters, optimizing a complete honeycomb structure can control the specific surface area, density, pore size, appearance, and other morphological characteristics of GCA, and further change its mechanical and thermal properties and electrical conductivity. 

### 3.2. Mesoporous Core–Shell Structure of GCA

Microspheres with mesoporous and core–shell structure produced by GCA exhibit greater strength and have wider application. These aerogels can usually be obtained using the sol-gel, blending, or other method to obtain hydrogels, which are then transformed into aerogel by freeze-drying or super-zero boundary CO_2_ drying. Liu et al. [48] prepared GCA with well-shaped microspheres and with internal honeycomb, which could be exploited as a photocatalyst and adsorbent. As shown in Figure 14, the shape of the honeycomb and the mesoporous microspheres with an internal radial microchannel structure help in shortening the diffusion path of pollutants and promoting a rapid adsorption equilibrium in the treatment of sewage. Therefore, the relationship between the design of the honeycomb structure and the performance and application of GCA needs further study.

Compared with the mesoporous microspheres structure, the core–shell structure is more complicated. Researchers have adjusted the pore structure of GCA aerogels by controlling the freezing temperature and direction of ice crystals formed between graphene layers. In addition, in order to form dense nuclei and sparse filling shells, the freezing temperature gradient is needed to control the nucleation and growth of ice crystals to obtain GCA microspheres with core–shell structure (Figure 15). This unique structure achieves high compressive strength by continuously distributing mechanical loads between core and shell, resulting in improved mechanical, electrical, and thermodynamic properties. 

### 3.3. Methods for Adjusting the Porous Structure of GCA

The pore structure of GCA can be adjusted by controlling the concentration of polymers, inorganic nanomaterials, and graphene nanosheets. The template effects and self-assembly process of graphene-based materials can be used to change the pore structure of aerogels and other materials [50]. In addition, the newly developed impregnation method and 3D printing technology have better effects on regulating the pore structure. Compared with traditional strategies, they greatly broaden the unique functional structure with controllable structural parameters [51]. 

## 4. Properties of GCA

### 4.1. Adsorptive Properties

Adsorption properties include physical and chemical adsorption. GCA aerogel’s abilities are connected with the structural design and performance of the material. In terms of structural design, 3D GCA usually has a larger specific surface area, higher porosity, and an interconnected porous structure, which can increase its combination with other functional materials and promote the diffusion of ions and molecules [52]. In addition, magnetic inorganic nano-GCA (Co_3_O_4_, Fe_3_O_4_) is easy to recycle [53]. Introducing different functional groups, such as amino and phosphate groups, into 3D GCA leads to different adsorption mechanisms, such as ion exchange, complexation, chelation, electrostatic interaction, hydrogen bond, π–π, hydrophobic interaction, etc. [54]. 

The adsorption of metal ions requires a large number of oxygen-containing functional groups (–OH, –COOH, –O–) on the GO surface combined with positively charged heavy metal ions through electrostatic interactions, or surface complexes to form metal complexes [55]. At the same time, a porous network structure effectively prevents the accumulation of GO, promotes the free diffusion of heavy metal ions, and broadens the contact probability of heavy metal ions with an active center. Additionally, π–π interactions and hydrogen bonds were observed between the edges of graphene [56], which efficiently bound and adsorbed dye molecules and metal ions (Figure 16) [47]. 

Furthermore, after the design and construction of roughness, the introduction of hydrophobic groups, and the thermal annealing reduction process, GCA was shown to have hydrophobic and lipophilic adsorption properties, which benefited oil–water separation [57]. Using the unique 3D porous structure, hydrogen bonds, and π-π interaction characteristics also adsorbed antibiotics and other drugs. These adsorption characteristics make it valuable for application in sewage purification.

### 4.2. Mechanical Properties

The key mechanical properties of GCA include compressive strength and deformation resistance. However, pure graphene aerogel is too weak for practical use. For this reason, various cross-linking agents or structural enhancers are introduced to prepare GCA with high elasticity and mechanical properties [58]. For example, some 2D materials were used as structural reinforcement materials for GCA, such as MXenes, which helped the aerogel obtain a super-elastic structure [59]. Organic nanomaterials, synthetic polymers, and natural polymers are commonly used as mechanical reinforcements for GCA to maximize its mechanical performance [60]. 

As shown in Figure 17, cross-linking agents or structural strengthening materials improved the mechanical stability and the relationship between microstructure anisotropy and mechanical strength of the transverse (TD) and longitudinal (LD) direction of composite aerogels; compared with pure graphene, the maximum compressive strength of LD and TD was increased by more than 10 times (10–50 kPa) [61]. Therefore, reasonable design of the preparation process is very important in order to improve the mechanical properties of composite aerogels without adversely affecting other functional performance. 

### 4.3. Thermal Properties

The thermal conductivity of graphene is 5000 W·m^−1^·K^−1^, which makes it a suitable filler for the preparation of composites, with excellent thermal properties [62]. The inherent characteristics of aerogels, such as light weight and very low thermal conductivity, provide new ideas for the development of heat insulation materials. However, the distribution of graphene nanosheets, the density of aerogel, and the pore structure of GCA affect its thermal properties; for example, uniformly distributed graphene nanosheets can greatly reduce or eliminate the contact thermal resistance and thus improve the thermal conductivity, whereas the low bulk density of the aerogel decreases thermal conductivity [63]. Therefore, in order to design materials with better thermal properties, the advantages of both need to be taken into account. For example, Wang et al. [64] prepared layered porous and continuous silk fibroin SF/GCA fibers by wet-spinning and freeze-drying, showed that the introduction of GO not only improved the mechanical properties, but also significantly raised the thermal properties under infrared radiation. Compared with pure SF aerogel fibers, the surface temperature of the SF/GCA was increased by 2.6 °C after infrared radiation for 30 s. At the same time, layered porous and hollow fiber structures reduced heat conduction and inhibited thermal radiation, providing good thermal insulation of SF/GCA fibers (Figure 18). 

In short, the introduction of graphene-based materials improves the thermal performance of aerogel and provides a scheme to overcome the problem of energy consumption [65], which gives it great application potential in thermal insulation materials, flame retardant materials, and electronic devices.

### 4.4. Electrochemical Properties

The ultra-high electrical conductivity of graphene is one of its most attractive characteristics, which can reach 10^7^–10^8^ S/cm, and makes it possible to prepare GCA with excellent electrical performance [66]. However, due to the contact resistance between graphene nanosheets, the electrical conductivity of GCA usually decreases significantly. There are two common methods to further increase the electrical properties of graphene aerogels. The first method is to introduce conductive polymers or doping metal oxides into the aerogel, and the synergistic effect between components can improve the electrical conductivity and structural stability of the polymer/GCA. The second method is to uniformly coat conductive polymer/GA materials on the substrate by spraying or spinning [67], which will significantly improve the conductivity of the aerogel as a supercapacitor and energy storage material.

Common metal oxides such as Fe_2_O_3_, Co(OH), Co_3_O_4_, MnO_2_, and MoS_2_ increase the electrical conductivity of GCA. For example, with the hydrothermal method, GCA was obtained by doping nano-Fe_2_O_3_ into graphene aerogel, and its specific capacitance was 81.3 F·g^−1^ at a constant current density of 1 A·g^−1^ and working potential of −0.8–0.8 V. GCA can also be functionalized by O, N, S, B, etc. [68]. Yun et al. [69] doped N into carbon quantum dots (CQDs) and then combined them with rGO and different ratios of Fe_2_O_3_ to produce GCA by form N-CQDs/rGO/Fe_2_O_3_ composite, and the preparation process is shown in Figure 19. The composite aerogel had excellent electrochemical performance and ultra-high specific capacity due to its high surface area and porous layered structure, as well as the synergistic effect of high-capacity Fe_2_O_3_ and stable high-conductivity N-CQD/rGO. The proper doping proportion can further improve electrical conductivity [70]. Yang et al. [71] successfully prepared 3D MXene/rGO composite aerogel by the ice template method and coating with carboxylated polyurethane (PU). This not only had excellent electrochemical performance, but also good self-healing ability, and the capacitance retention rate reached 91% in 15,000 cycles, providing a method for use in long-life multi-function electronic devices. Therefore, the preparation of GCA with excellent electrochemical performance, whether coated with polymer conductive materials or doped with inorganic nanoparticles or multicomponent composites, mainly depends on the filler ratio, the inherent electrical conductivity of the graphene-based nanosheets and materials used, and the control of the micro-morphology.

## 5. Application of GCA

### 5.1. Adsorption Removal of Contaminants from Water

Adsorption is a popular sewage treatment because of low cost, simple operation, large adsorption capacity, and high removal rate [72]. GCA is an ideal adsorbent for sewage treatment owing to its higher adsorption capacity and easy reuse [73]. Zang et al. found that more porous CS/GCA had good adsorption and removal effects for Pb^2+^. When the GO content in the aerogel was 5 wt%, the adsorption capacity of Pb^2+^ increased from 68.5 to 100 mg·g^−1^. The adsorption mechanism of common metal ions Pb^2+^ and Cu^2+^ may consist of intergroup coordination and complexation (Figure 20) [74]. 

The surface properties of the adsorbent and the chemical properties (structure, hydrophobicity, polarity) of the aerogels [75] determine the types of pollutants that can be adsorbed [76]. Preparation of GCA with high hydrophobicity and high specific surface area provides a feasible solution for the removal of organic oil pollution. Yang et al. [77] introduced fluoroalkyl-silane into GCA by gas–liquid deposition to obtain superhydrophobic graphene aerogels (SGAs) with super hydrophobicity, super lipophilicity, ultra-low density, large specific surface area, excellent adsorption capacity, and adsorption recycling (Figure 21), which has great potential in the field of oil–water separation.

Table 3 lists the studies on the adsorption properties of GCA for heavy metals, organic compounds, dyes, and other pollutants. In terms of recyclability, most studies have shown that recovery and desorption efficiency were improved by selecting suitable desorbents (acids, alkalis, and organic solvents), and the 3D structure of GA had a unique effect in this regard [78]. This shows that GCA has broad prospects in adsorption treatment of water pollution.

### 5.2. Application of Sensors and Supercapacitors

The unique porous structure of GCA gives it good flexibility and elasticity, and it has become the preferred material for piezoelectric resistance sensors. In addition, GCA has great application potential in the field of energy storage and sensing, including supercapacitors, lithium batteries, solar cells, and fuel cells, because of its high conductivity, high electrochemical stability, and good mechanical stability [91]. 

Pressure sensitivity plays a key role in piezoelectric resistance materials, which determines the sensitivity of resistance materials. Generally, a highly sensitive aerogel can be obtained by controlling the composition of the GCA. Wei et al. [92] used borate as a reinforcing material in a 3D graphene aerogel structure to obtain nitrogen and boron co-modification (NBGC) aerogel with excellent elasticity and compressibility and good electrochemical properties. This kind of aerogel provided a fast external stress change current response with specific capacitance up to 336 F·g^−1^; when the stress increased from 0.05 N to 10 N, the response current varied from 0.44–2.89 mA to 10 N, so it could be applied in high-performance stress sensors (Figure 22). 

The application of GCA in energy storage is shown in Table 4. The introduction of polymer or inorganic nano-active materials into the aerogel structure improves the electrochemical energy storage of super capacitors [93]. For example, using high-conductivity Cu/Cu_x_O to modify the rGO network structure, C aerogel with high apparent conductivity was obtained [94], which was two to three orders of magnitude higher than pure graphene aerogel (0.1 S/m). Li et al. [88] designed a new type of PPy layer coated sulfur/GCA by vapor deposition, which was used as the cathode of a lithium–sulfur battery. PPy as a uniform coating ensures long-time, stable cycle performance of lithium–sulfur batteries, and it also has excellent electrochemical properties, such as high specific capacity. The discharge capacity at 0.2 C after 500 cycles reached 1167 mAh·g^−1^ after 500 cycles. In addition, Table 4 also shows that the electrochemical energy storage and conductivity of GCA doped with heteroatoms N or B are greatly improved, and the active specific surface area is increased. The synergistic effect of N and B co-doping promotes the charge transfer between adjacent carbon atoms, improves the electrochemical performance of carbon-based materials, and gives them excellent power density and charge–discharge rate, which makes this a very promising super energy storage capacitor material [95]. This shows that 3D GCA has great research value with regard to energy storage materials. In the future, designing GCA with a porous structure and a larger specific surface area, while maintaining a good conductive path for effective charge transfer, is a problem that researchers will need to pay attention to.

### 5.3. Application of Heat-Insulation and Flame-Retardant Materials

Aerogels have unparalleled advantages as thermal insulation materials. The reason is that the ultra-high porosity of aerogel reduces heat conduction, and the pore walls in the aerogel network can effectively restrain thermal radiation. When the aerogel has smaller pore size, thermal convection will be reduced [104]. Some polymer aerogels, such as PVA [105], cellulose [106], and pectin [107] aerogels, are excellent thermal insulation materials, but their application is limited because of poor thermal stability and flame retardancy. Fortunately, carbon nanomaterials such as graphene materials can improve the physical and thermal properties of polymer aerogels. Because the density, porosity, and complex skeleton structure have an influence on the porous thermal insulation materials, composite aerogel with low density, low thermal conductivity, and high strength prepared by quartz fiber and nitrogen-doped graphene has great application potential in aviation for thermal insulation. 

In terms of flame-retardant materials, highly thermal stable graphene aerogel with large porosity eliminates heat rapidly during combustion [108]. Hence, taking advantage of the flame retardancy of graphene aerogel and the low thermal conductivity of composite, aerogels prepared with phenolic resin have ultra-low thermal conductivity, high thermal stability, and good flame retardancy [109]. Ceramic fillers with Al_2_O_3_ ceramic and graphene have been designed with a layered honeycomb microstructure, showing a coupling strengthening effect between the graphene skeleton and the Al_2_O_3_ ceramic nanolayer. The composite aerogel not only has ultra-light density, reversible compressibility, and high electrical conductivity, but also great application prospects in flame retardancy, thermal insulation, and so on [110]. 

### 5.4. Biomedical Applications

In the biomedical field, some degrees of biocompatibility, biodegradability and antibacterial properties of graphene-based materials are beneficial [111]. The toxicity of graphene-based materials is related to many aspects such as content, size, surface chemistry, cell lines, morphologies, administration route, etc., and this is also applicable to similar GCA studies [112]. For example, in the study of GCA used in in situ bone regeneration, the content of nano graphene-based materials can affect the biocompatibility and biodegradability of GCA, and the content of GO is 0.10% aerogel is more conducive to cell increment [113]. Nanoscale small-sized GO has a lower level of cytotoxicity so that it can be used as a drug delivery carrier; the graphene aerogel nanoparticles (GANPs) were prepared as drug delivery carriers with high pH sensitivity, and released after 5 days in vivo, and are expected to be used in nanomedicine in the future [114]. The toxicity of graphene-based materials is different in vivo and in vitro, which is affected by their physical and chemical properties, such as functional groups, charges, sizes, stiffness, hydrophobicity and structural defects [112]. Luo et al. synthesized three-dimensional multi-functional GCA materials using tannic acid as raw material, which showed high porosity, low density, good hydrophobicity, good mechanical properties, high thermal stability, strong antibacterial properties, and sterilization rates of 58.12 and 99.99% for *Escherichia coli* and *Staphylococci*, respectively [115]. Therefore, a large number of studies in the biomedical field have focused on its safety and reducing its cytotoxicity, such as through the introduction of some biocompatible materials such as chitosan, collagen, gelatin, serum albumin and so on [116]. In addition, graphene-based nanosheets are used as structural reinforcement materials to prevent cells from collapsing during growth, thus stabilizing the three-dimensional structure, which is more obvious in the case of good dispersion of nanoparticles and good affinity of polymer fillers. For example, a thin biocompatible coating can be formed on the surface of three-dimensional graphene, which can be used as a high-strength biocompatible scaffold material for nervous system regeneration and musculoskeletal tissue engineering. This presented good tissue integrity renewal ability and inhibition of lesion expansion after spinal cord injury [117].

Furthermore, these materials could also be applied as biosensors to provide a wealth of information for early diagnosis of diseases and prevention of their evolution [118]. Composite material modified with 3D GCA and nano-CuO, with high sensitivity, was exploited as a biosensor for the detection of ascorbic acid [119]. Graphene aerogel was shown to detect dopamine with high sensitivity (619.6 μA·μM^−1^·cm^−2^), which was attributed to the highly conductive 3D multichannel, high charge transfer rate, and efficient transport guaranteed by the close interaction between dopamine and graphene [120]. The multifunctional silica/GCA material can be used as a biosensor for the detection of insulin (INS) with high selectivity and sensitivity [121]. The extensive research on GCA in the biomedical field is focused on its convenience as a drug carrier, antibacterial, biological scaffold, sensor, and so on. Designing GCA material with flexible biological and intelligent properties in the future will broaden its application in the biomedical field. Research on GCA in biomedicine has great potential, and is worth promoting further.

## 6. Conclusions and Outlook

This paper reviews the material composition, preparation methods, structural characteristics, properties, and applications of GCA. Here we focus on several types of GCA, including graphene-based/2D nanomaterial (MXenes) aerogel for sensors and supercapacitors, graphene-based/inorganic nanomaterial (SiO_2_, SnO_2_, SnO_2_, TiO_2_) aerogel and heat-insulating flame-retardant materials, and graphene-based/synthetic polymer aerogel and graphene-/natural sugar-based polymer aerogel used as adsorbents to remove metal ions and dye contaminants in water. 

In addition, this paper also describes the influence of differences in the morphological structure (specific surface area, density, pore size, etc.) of GCA on its mechanical, thermal, and electrochemical properties. At the same time, the commonly used methods for preparing GCA are briefly described, including template method, self-assembly, chemical vapor deposition, 3D printing, and performance improvement strategies (doping, coating, cross-linking). GCA not only makes up for the deficiencies in the mechanical properties of graphene aerogel, but also retains excellent electrical conductivity, good mechanical flexibility, low thermal expansion coefficient, and other physical properties. It meets the special requirements for material properties in new fields, such as environmental purification, sensing, energy storage, biomedicine, heat insulation, and fire protection, and has become a hot spot in the field of graphene research in recent years. However, at present, the preparation method of graphene aerogel is relatively cumbersome, and it is necessary to find a quick preparation method to realize large-scale industrial production, although the chemical exfoliation method described in previous studies can be used to easily prepare graphene and its derivatives on a large scale. This still needs to be optimized to control the structure and size of materials. In terms of well-designed GCA, it is necessary to further explore the relationship between its properties and microstructure and optimize the preparation process parameters. For example, in the previous introduction, the pore structure of the aerogel was adjusted by controlling the formation of ice crystals between graphene sheets, but it is difficult to obtain uniform pore structure due to the existence of a temperature gradient, so it is still a difficult problem to skillfully control the morphology and size distribution of the product. In addition, most of the research has remained in the laboratory stage. Researchers should combine studies with production demand and broaden the scope of application.

In summary, the future research direction of GCA will move toward the design and preparation of new graphene derivatives and their composite materials, with multi-dimensionality and better performance. Compared with GCA, it is difficult to use other materials in so many areas, especially energy storage, sensing, and adsorption, which are based on the characteristics of graphene-based and aerogel materials. With the continuous advancement of science and technology, GCA that is more environmentally friendly and has excellent application performance and better structural properties will be prepared in the future, which is both an opportunity and a challenge.

## Figures and Tables

**Figure 1 materials-15-00299-f001:**
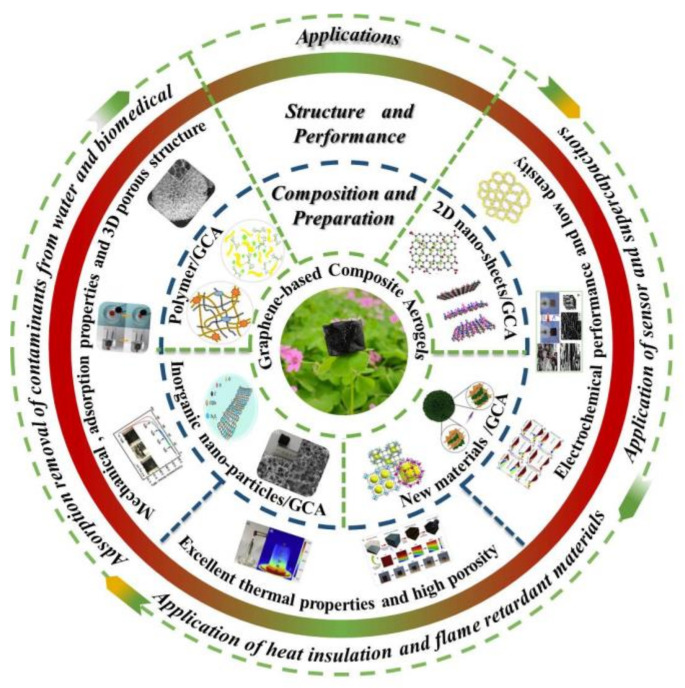
Composition, structure, properties, and applications of GCA.

**Figure 2 materials-15-00299-f002:**
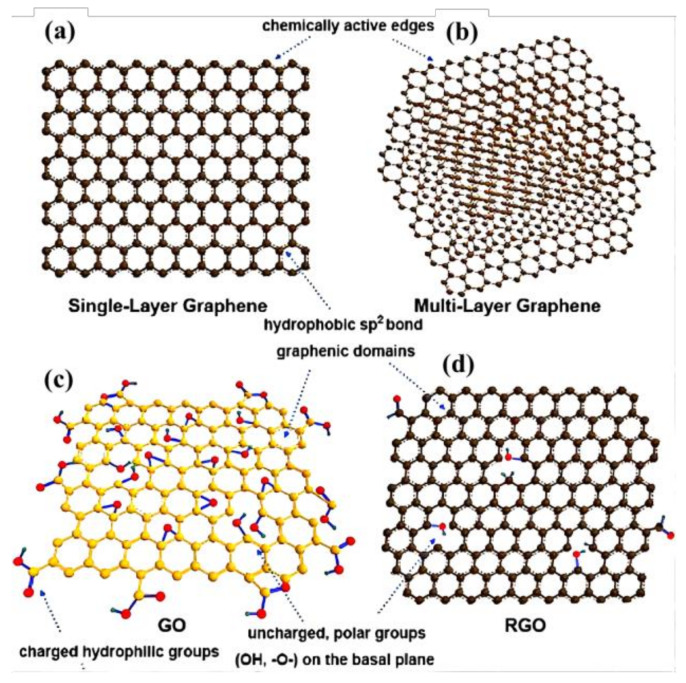
Structural schematic of graphene-based materials: (**a**) single-layer graphene, (**b**) multi-layer graphene, (**c**) GO, (**d**) rGO [6].

**Figure 3 materials-15-00299-f003:**
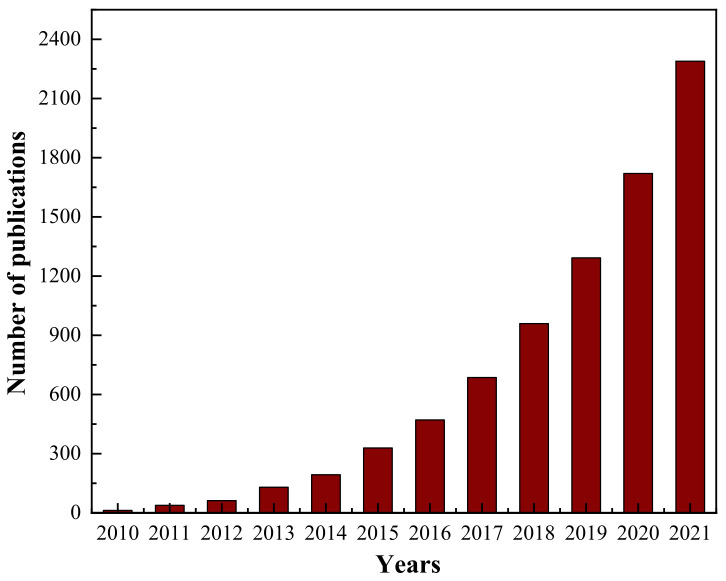
Number of papers on GCA in the last 12 years (as of 20 September 2021 by Science Direct record).

**Figure 4 materials-15-00299-f004:**
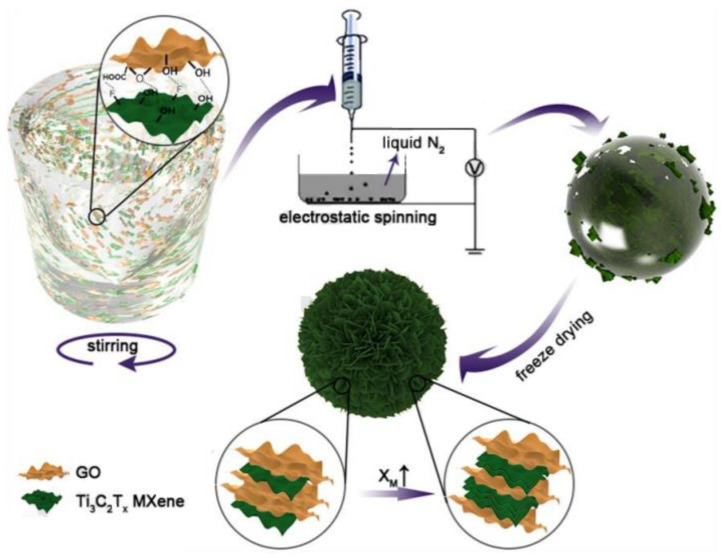
Assembly mechanism of MXenes@GO composite aerogel [10].

**Figure 5 materials-15-00299-f005:**
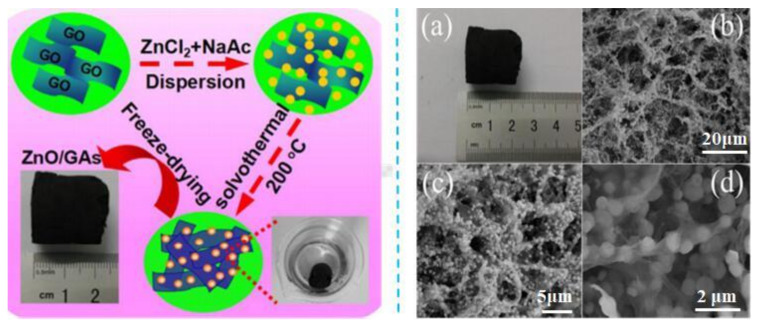
Preparation process of ZnO/GCA: (**a**) Macro-morphology and (**b**–**d**) Micro-morphology by SEM [14].

**Figure 6 materials-15-00299-f006:**
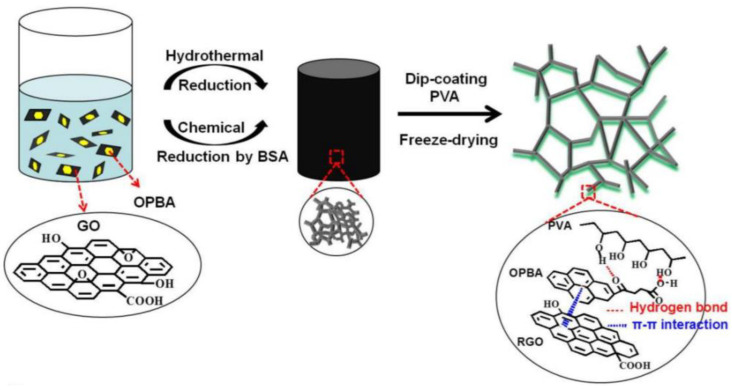
Schematic diagram of preparation process of PVA/GCA [17].

**Figure 7 materials-15-00299-f007:**
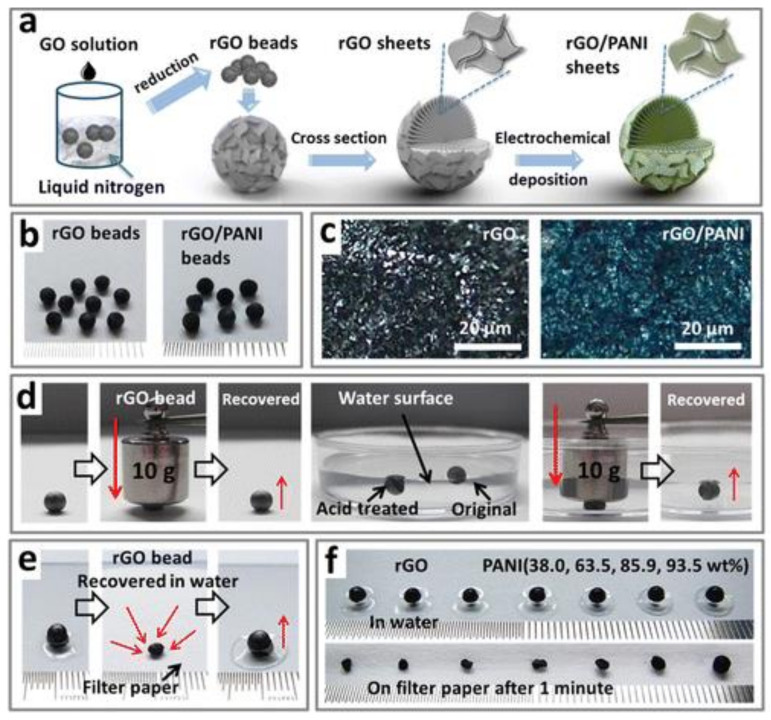
Schematic diagram of preparation of PANI/GCA: (**a**) Freeze-casting to prepare PANI/rGO aerogels; (**b**) PANI/rGO aerogel samples; (**c**) Microstructure of PANI/rGO aerogels; (**d**) Preparing rGO aerogels; (**e**) Recovery of shrunken rGO aerogels in water; (**f**) Shrinkage degree of GO and PANI/rGO aerogel by loss of water [19].

**Figure 8 materials-15-00299-f008:**
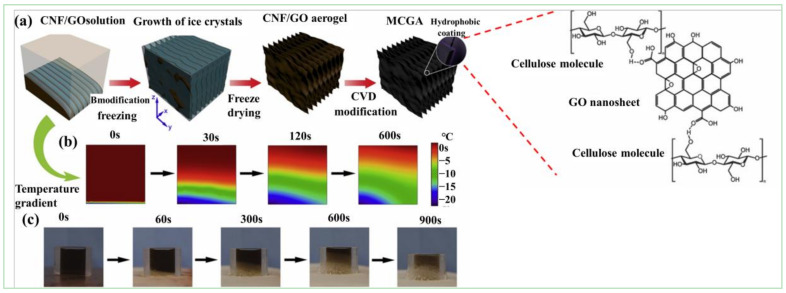
Schematic of hydrogen bond interactions between GCA and cellulose [34]: (**a**) Bi-directional freeze-drying preparation of cellulose/GCA; (**b**) Temperature gradient simulation in the freezing process; (**c**) Digital image of solution freezing process of two-way freezing [21].

**Figure 9 materials-15-00299-f009:**
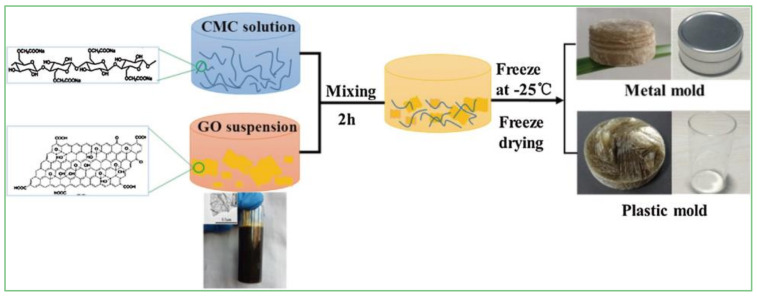
Schematic demonstration of CMC/GCA composite aerogel preparation [22].

**Figure 10 materials-15-00299-f010:**
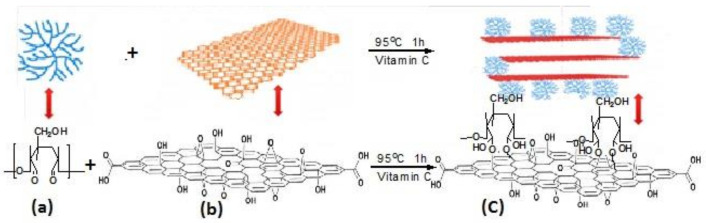
Fabrication and structure of starch/GCA [23]. (**a**) Starch; (**b**) Graphene oxide; (**c**) Starch/GCA.

**Figure 11 materials-15-00299-f011:**
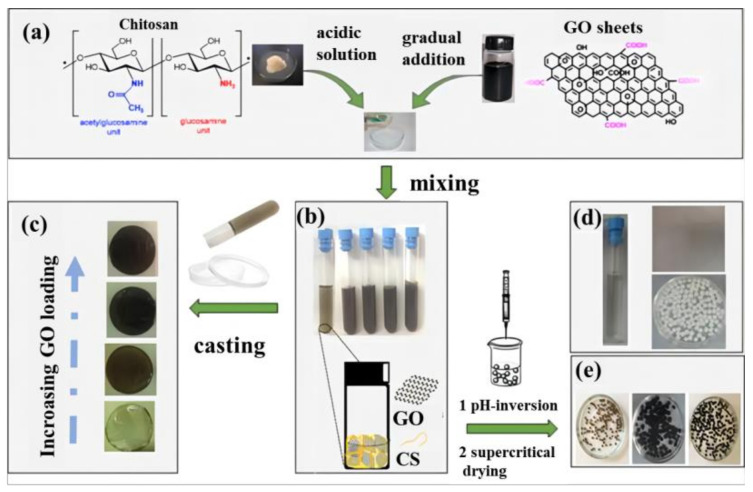
Schematic diagram of preparation process of CS/GCA [38]. (**a**) Chemical structure of chitosan and graphene oxide as two mixed dissimilar phases; (**b**) Digital photo of CS-GO acidic-aqueous solution with increasing GO weight percentage; (**c**) The as-prepared hybrid films (denoted as CS-GO) with increasing GO amount; (**d**) Self-standing chitosan-graphene oxide aerogel microspheres with increasing amount of GO in the mixture; (**e**) Blank materials prepared for comparison: chitosan aqueous acidic solution, chitosan film and chitosan microspheres.

**Figure 12 materials-15-00299-f012:**
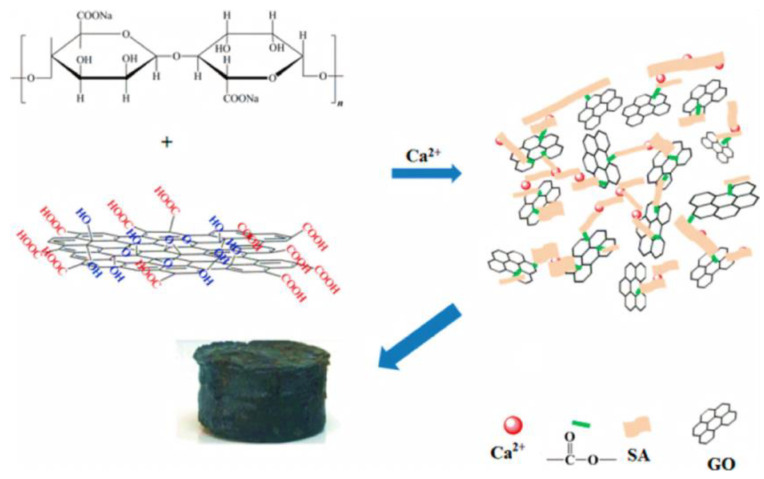
Synthesis route of SA/GO composite aerogel [39].

**Figure 13 materials-15-00299-f013:**
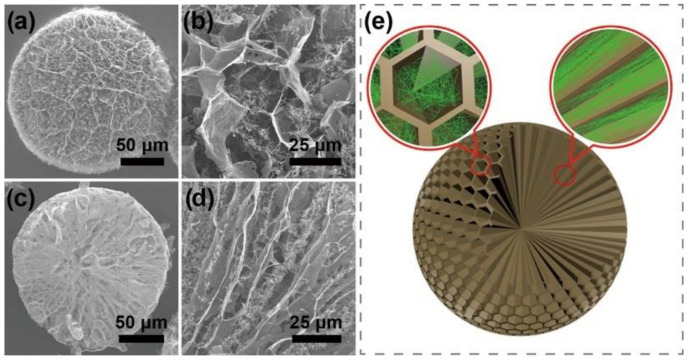
SEM images of: (**a**,**b**) Graphene aerogel, (**c**,**d**) GCA, and (**e**) Microchannel structures of GCA [47].

**Figure 14 materials-15-00299-f014:**
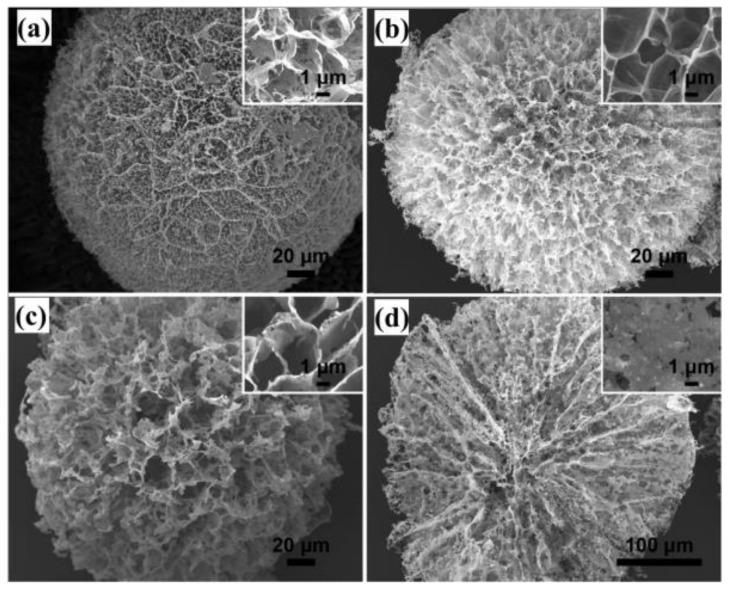
SEM images of: (**a**) GCA, (**b**) GCA1 (50% Ag_3_PO_4_); (**c**) GCA3 (75% Ag_3_PO_4_); (**d**) Cross-section of GCA microspheres [48].

**Figure 15 materials-15-00299-f015:**
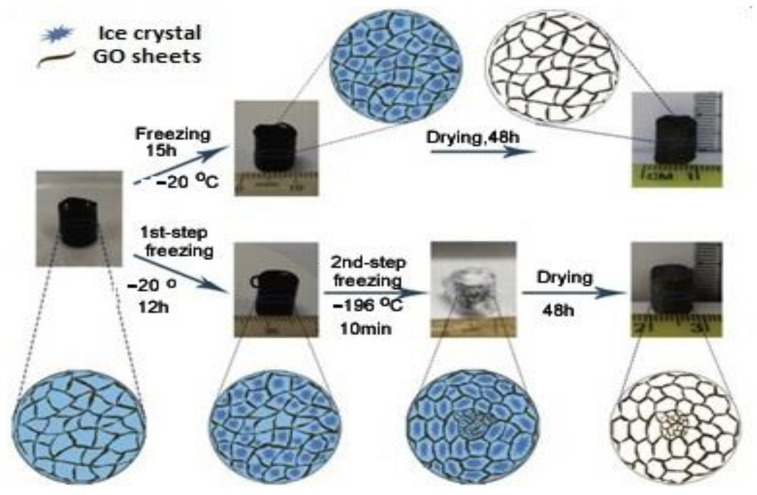
Schematic illustration of fabrication of GCA microspheres with core–shell structure [49].

**Figure 16 materials-15-00299-f016:**
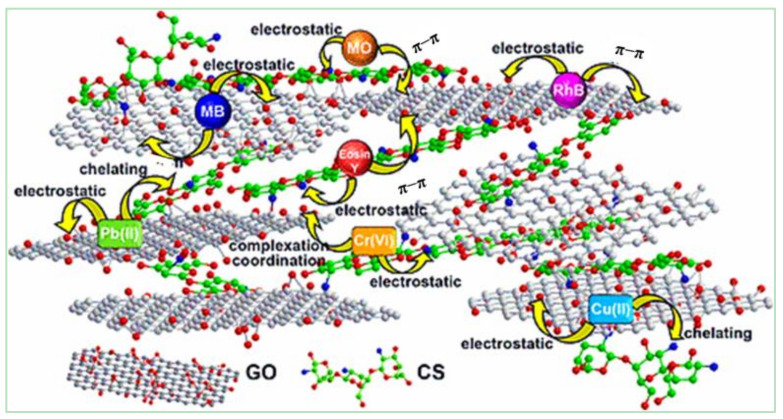
Schematic diagram of purification mechanisms of CS/GO aerogels for sewage [47].

**Figure 17 materials-15-00299-f017:**
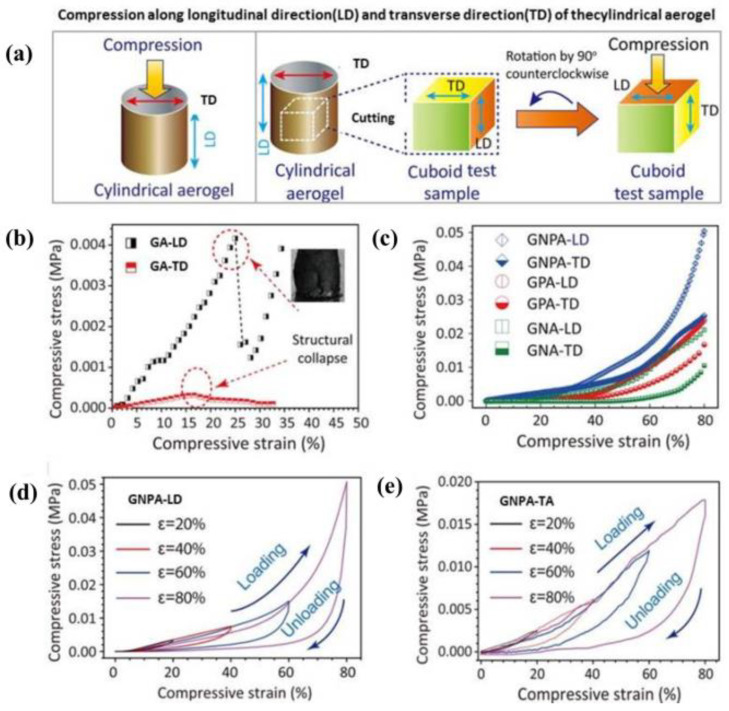
(**a**) Schematic of sampling instructions for expression test along TD and LD; (**b**,**c**) Stress–strain surveys of all testing samples in LD and TD; (**d**,**e**) Loading–unloading stress–strain surveys for GCA-LD and GCA-TD under different strains [61].

**Figure 18 materials-15-00299-f018:**
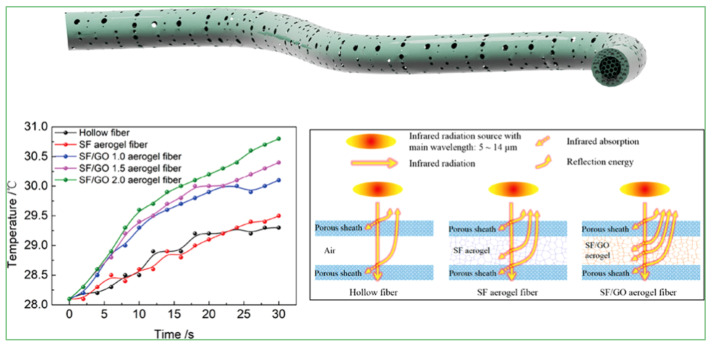
Radiative heating properties of SF/GCA fibers [64].

**Figure 19 materials-15-00299-f019:**
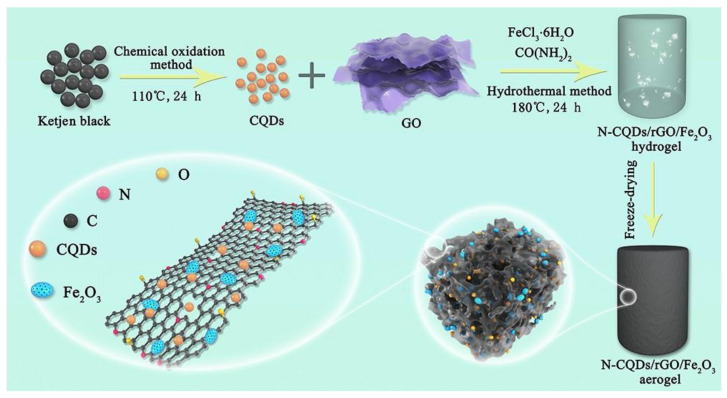
Schematic demonstration of GCA aerogel synthesis [67].

**Figure 20 materials-15-00299-f020:**
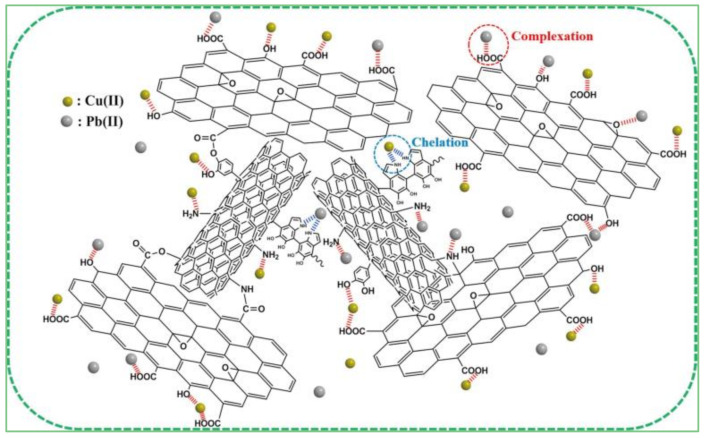
Complexation (red) and communication (blue) interactions of Pb (II) and Cu (II) adsorption on GCA [74].

**Figure 21 materials-15-00299-f021:**
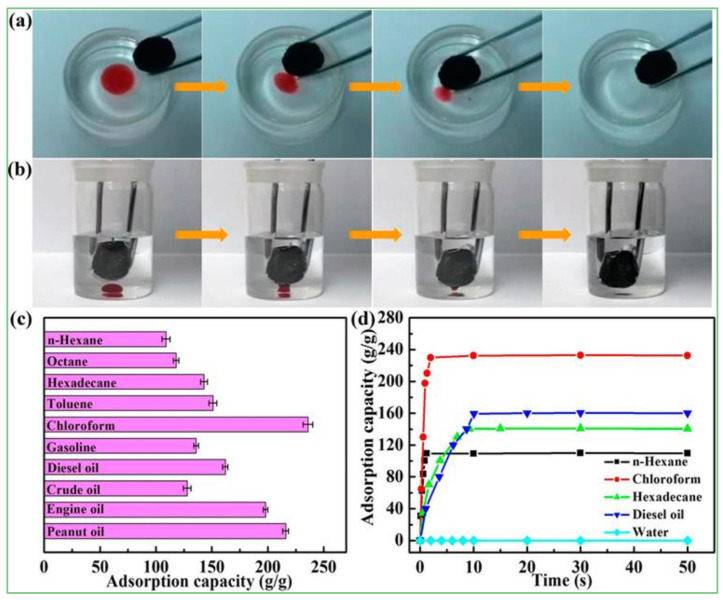
Adsorption performance of SGAs for oil. (**a**) Snapshots showing the SGAs adsorbs hexadecane (dyed by oil red) floating on the water; (**b**) Snapshots showing the SGAs adsorbs chloroform (dyed by oil red) from the bottom of the water; (**c**) The adsorption capacity of the SGAs for various kinds of organic solvents and oils; (**d**) The time-dependent sorption behaviour of various oily compounds and water by the SGAs [77].

**Figure 22 materials-15-00299-f022:**
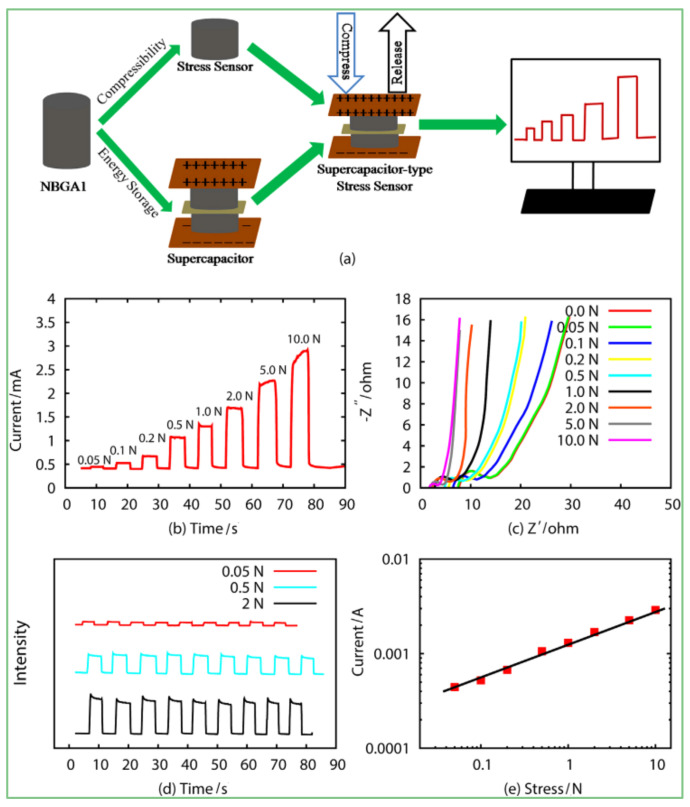
Electrochemical and pressure response performance of NBGC aerogel as supercapacitor and stress sensor. (**a**) A schematic illustration of assembled NBGC aerogel as a supercapacitor stress sensor; (**b**) The response currentetime (Iet) characteristic curves of the NBGC aerogel stress sensor at different stress (T) of 0.05e10 N; (**c**) The EIS of the cell under different stress; (**d**) The stress sensor cycle stability of device at different stress of 0.05 N, 0.5 N and 2 N, respectively; (**e**) The relationship between the response current and the stress. [92].

**Table 1 materials-15-00299-t001:** Preparation methods and applications of common GCA materials.

GCA	Preparation Principle	Applications	Reference
2D nanosheets/GCA	3D printing	Energy storage materials	[9]
	Electrospinning	Wave-absorbing materials	[10]
	Self-assembly	Electrocatalysts	[11]
Inorganic nanoparticle materials/GCA	Self-assembly	Wearable piezoresistive sensors	[12]
	Sol-gel method	Energy storage material	[13]
	Sol-gel method	Gas sensing materials	[14]
	CVD	Adsorption materials	[15]
	Self-assembly	Catalytic material	[16]
Synthetic polymer/GCA	Sol-gel method	Pressure response sensor	[17]
	Template method	Thermal insulation materials	[18]
	Immersion method	Porous electrode	[19]
	Self-assembly	Energy storage material	[20]
Natural polymer/GCA	CVD	Adsorption materials	[21]
	Template method	Thermal insulation materials	[22]
	Self-assembly	Supercapacitor materials	[23]
	Sol-gel method	Adsorption materials	[24]
	Chemical cross-linking	Adsorption materials	[25]

**Table 2 materials-15-00299-t002:** Fabrication methods and morphology of GCA materials.

Composite Aerogel Material	Fabrication Method	Specific Surface Area (m^2^·g^−1^)	Density (mg·cm^−3^)	Pore Volume (cm^3^·g^−1^)	Diameter (nm)	Reference
GNPA	Self-assembly	/	59.3	/	5	[12]
GO/SiO_2_	Sol-gel method	889	/	3.72	16.75	[40]
VO_2_/GA @NiS_2_	Sol-gel method	141.1	/	/	17.3	[41]
Co-N-GA	Self-assembly	485	0.29	0.71	/	[11]
PPy@GA	Self-assembly	686	7.8	/	/	[20]
MGGNA	Freeze-drying	45.1	2.32	/	/	[42]
GNR	Self-assembly	113.1	9.33	/	/	[43]
MoS_2_-RGO	3D printing	/	/	/	100–200	[9]
3D-GMOs	CVD	560	22	/	/	[15]
N-CMS/GA	Mixed	450	/	/	3.4–36.9	[44]

GNPA, graphene–PVA–co–PE nanofiber–PVA aerogel; Co-N-GA, hierarchically porous Co-N functionalized graphene aerogel; PPy@GA, poly-pyrrole nanosphere graphene aerogel; MGGNA, capillary-like hydrophobic GO/GONR-APTES composite aerogel; GNR, graphene nanoribbon aerogel; 3D-GMOs, high-density three-dimensional GA macroscopic objects; N-CMS/GA, nitrogen-doped carbon microsphere/graphene aerogel.

**Table 3 materials-15-00299-t003:** Adsorption properties of different GCAs.

Adsorbent	Contaminants	Adsorption Conditions	Isotherm Model	q_m_ (mg·g^−1^)	Reference
RGO/ZIF-67	CV MO	pH 3–9, T = 298 K, t = 0–16 h	Langmuir	1714 426	[79]
MWCNT-PDA	Cu^2+^ Pb^2+^	pH 2–7, T = 298 K, t = 0–10 h	Langmuir	318.47 350.87	[74]
3D-Fe_3_O_4_/GA	As^5+^	pH 7, T = 298 K, t = 0–16 h	Langmuir	40.05	[80]
3D-SRGO	Cd^2+^	pH 2–9, T = 298 K, t = 0–24 h	Langmuir	234.8	[81]
GA/SiO_2_	Hg^2+^	pH 2–10, T = 296 K, t = 0–1.5 h	Langmuir	500.0	[82]
3D δ-MnO_2_	Pb^2+^ Cd^2+^ Cu^2+^	pH 2–6, T = 298 K, t = 0–3 h	Langmuir	643.62 250.31 228.46	[83]
3D GO/SA	MB	pH 4–9, T = 293 K, t = 0–24 h	Langmuir	4.25	[84]
PAM/GO	Magenta	pH 2.6–8.9, T = 303 K, t = 0–55 h	Langmuir	1034.3	[85]
GO-AL	MB	pH 3.0, T = 303 K, t = 0–4 h	Langmuir	1185.98	[86]
PPGA	N-hexane, MO	pH 3.0, T = 303 K, t = 0–4 h	Langmuir	63.5 202.8	[87]
RGO/REMO	RhB,	t = 0–24 h	Langmuir	243.4	[88]
GAS-MS	phenol, catechol, resorcinol, hydroquinone	pH 3.0, T = 298 K, t = 24 h	Langmuir&Freundlich	90 66 22 67	[28]
CNF/GO	Tetracycline	pH 2–12, T = 298 K, t = 24 h	Langmuir	454.6	[89]
3DG	Methyl bromide	pH 7.5, T = 298 K, t = 0–5 h	Langmuir	685	[90]

RGO/ZIF-67, three-dimensional rGO/zeolitic imidazolate framework-67 aerogel; MWCNT-PDA, graphene/polydopamine modified multiwalled carbon nanotube hybrid aerogel; 3D-SRGO, 3D sulfonated reduced GO aerogel; 3D δ-MnO_2_, 3D graphene/delta-MnO_2_ aerogel; GO-AL, GO/alkali lignin aerogel composite; PPGA, polydopamine and poly-ethylenimine co-functionalized GO aerogel; RGO/REMO, RGO/rare earth–metal oxide aerogel; GAS-MS, 3D graphene aerogel–mesoporous silica; CNF/GO, cellulose nanofibril/graphene oxide hybrid aerogel; 3DG, three-dimensional graphene aerogel.

**Table 4 materials-15-00299-t004:** Energy storage performance of different GCA materials.

Composite Aerogel Materials	Specific Capacitance (F·g^−1^)	Energy Density (Wh·kg^−1^)	Cyclic Behavior	Electrolyte	Reference
F-Fe_2_O_3_@MGA	1119@1 Ag^−1^	800	98.9% after 2000 cycles	3M KOH	[96]
LMP/rGO	4645@0.5Ag^−1^	11.79	91.2% after 10,000 cycles	6M KOH	[97]
PANI/CRGO/Co_3_O_4_	1247@1 Ag^−1^	190	92% after 3500 cycles	6M KOH	[90]
PPy/CRGO	304@0.5 Ag^−1^	/	58% after 50 cycles	6M KOH	[98]
GR-CNT	375@1 Ag^−1^	/	94.8% after 5000 cycles	6M KOH	[99]
CA	467@20 Ag^−1^	22.75	90.9% after 10,000 cycles	1M KOH	[100]
N,S-MGA	4929@2 Ag^−1^	686.7	98.7% after 5000 cycles	1M KOH	[101]
SnO_2_-GA	541@5 Ag^−1^	160	97.3% after 10,000 cycles	3M KOH	[102]
MnO_2_/P-RGO	645@1 Ag^−1^	59.2	94.6% after 10,000 cycles	1M Na_2_SO_4_	[103]

F-Fe_2_O_3_@MGA, flower-like Fe_2_O_3_@ multiple graphene aerogel; LMP/rGO, LiMnPO_4_/reduced GO aerogel; PANI/CRGO/Co_3_O_4_, self-assembled graphene/polyaniline/Co_3_O_4_ ternary hybrid aerogel; PPy/CRGO, conductive graphene/poly-pyrrole hybrid aerogel; GR-CNT, nitrogen-doped carbon aerogel; CA, nitrogen-doped carbon aerogel; N,S-MGA, nitrogen and sulfur-functionalized multiple graphene; MnO_2_/P-RGO, phytic acid modified manganese dioxide/GCA.

## Data Availability

Publicly available datasets were analyzed in this study. This data can be found here: https://www.webofscience.com/wos/alldb/basic-search (accessed on 25 November 2021).

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
