# Peer review of "Fabrication, Structure, Performance, and Application of Graphene-Based Composite Aerogel"

_materials, 2021, doi:10.3390/ma15010299_

Round 1

Reviewer 1 Report

The review reports the analysis of several articles related to graphene-based composite aerogel.

In my opinion, the article is not publishable in this form because of the following points.

  • There is the need for an accurate English revision. There are several errors all around the manuscript.
  • Which is the meaning of aerogel? This is the main point of the manuscript, but it's not clear. This explanation affects all studies reported. Indeed, it seems there are several studies reporting results on hydrogels, not properly aerogels.
  • What is GC? It is repeated in all text, without an explanation of the acronym.
  • Figure 1 is unreadable.
  • Figure 2: what is GA?
  • Table 1: please explain the meaning of diameter related to pore volume.
  • Table 1, Table 2 and Table 3 do not report only GO-based materials, but the analysis is extended to other ones.
  • Section 5.4 seems to be minimal, please remove or enlarge the overview.

Author Response

Response to Reviewer 1 Comments

English language and style

(x) Extensive editing of English language and style required
( ) Moderate English changes required
( ) English language and style are fine/minor spell check required
( ) I don't feel qualified to judge about the English language and style

Response :

We have sent this manuscript to the MDPI English Editing Center for grammatical and editing (English editing ID: English-38394), and we hope that the current English version of this manuscript can meet the needs of publication.  

Point 1: There is the need for an accurate English revision. There are several errors all around the manuscript.

 Response 1: We apologize for the mistakes in the manuscript and also carefully checked the entire manuscript for typographic, grammatical and formatting errors.

Point 2: Which is the meaning of aerogel? This is the main point of the manuscript, but it's not clear. This explanation affects all studies reported. Indeed, it seems there are several studies reporting results on hydrogels, not properly aerogels.

Response 2: Thank you for underlining this deficiency. (In other words, aerogel is a porous ultralight solid synthesized from a gel, in which the dispersion liquid is replaced with a gas, so the difference between aerogels and aerogels lies in their internal structure and density.) This section was revised and modified according to the information showed in the work suggested by the reviewer (Line 3, page 1).

Point 3: What is GC? It is repeated in all text, without an explanation of the acronym.

Response 3: We are grateful for the suggestion. To be more clear, we redefine it as GCA, which represents the graphene-based composite aerogel (Line 11, page 1)..

Point 4: Figure 1 is unreadable.

Response 4: We are extremely grateful to reviewer for pointing out this problem. We have introduced Figure 1 in more detail, which reveals the framework of the full text from the structure, performance and application of graphene-based composite aerogel.

Point 5: Figure 2: what is GA?

Response 5: Thank you for your comments, We used to use the abbreviation of GA for graphene, but now we have unified it in the full text.

 Point 6: Table 1: please explain the meaning of diameter related to pore volume.

Response 6: Thank you for your comment, and our reply is as follows: Pore volume is the volume of pore volume per unit mass of a porous solid and the pore volume is related to the inner diameter and outer diameter. When under the same inner diameter, the pore volume also decreases with the increase of the outer diameter of aerogels, and under the same outer diameter, the larger the inner diameter, the larger the pore volume.( Line 247, page9)

Point 7: Table 1, Table 2 and Table 3 do not report only GO-based materials, but the analysis is extended to other ones.

Response 7: Thank you for your precious comments and advice. Tables 1, Table 2 and Table 3 in the article include not only GO-based materials, but also rGO, graphene (GR), graphene aerogels (GA), multiple graphene aerogels, Nitrogen-doped carbon aerogels, etc., and other materials have been described in detail at the bottom of the table.

Point 8: Section 5.4 seems to be minimal, please remove or enlarge the overview.

Response 8: Thank you for underlining this deficiency. We have enlarged the overview of Section 5.4 and added several references. (Line 533, page 20).

Reviewer 2 Report

This paper shows a good review comparing and summarizing the use of application of graphene-based composite aerogel. There are some issues that need to address:

- I believe this review would benefit from a table that could compare and provide an overview of the discussed approaches. The table should include the advantages and limitations of each approach as well as findings.

- The language of the paper needs to be improved. There are some grammatical errors, please carefully check the whole manuscript.

- The introduction should be rewritten to show the highlights and novelty of the work. Also, authors can cite the following work in the introduction which is related to their work and recently reported:

- "Simulation of melting and solidification of graphene nanoparticles-PCM inside a dual tube heat exchanger with extended surface." Journal of Energy Storage 44 (2021): 103265.

- "Numerical study of the effect of graphene nanoparticles in calcium chloride hexahydrate-based phase change material on melting and freezing time in a circular cavity with a triangular obstacle." Journal of Energy Storage 43 (2021): 103243.

- section of drawbacks and future could be increased quality of the manuscript.

- Maybe at the beginning of the article, there should be a list of abbreviations?

- A review paper not only should summarize recently published works, but also should contain critical and comprehensive discussions. Therefore, check the writing for the whole manuscript. The review should not be presented by listing what has been done by others.

Author Response

English language and style

( ) Extensive editing of English language and style required
( ) Moderate English changes required
(x) English language and style are fine/minor spell check required
( ) I don't feel qualified to judge about the English language and style

Response: We have sent this manuscript to the MDPI English Editing Center for grammatical and editing (English editing ID: English-38394), and we hope that the current English version of this manuscript can meet the needs of publication.  

 Point 1: I believe this review would benefit from a table that could compare and provide an overview of the discussed approaches. The table should include the advantages and limitations of each approach as well as findings.

 Response 1: Our deepest gratitude goes to you for your careful work and thoughtful suggestions that have helped improve this paper substantially. We have created a new table in the description of the preparation method, and analyzed the advantages and disadvantages of the commonly used methods and a wide range of choices (Line 89, page 3). Due to the requirements of the length of the article, we also briefly analyzed and summarized the other tables in the article.

Point 2: The language of the paper needs to be improved. There are some grammatical errors, please carefully check the whole manuscript.

Response 2: We apologize for the language problems in the original manuscript. The language presentation was improved with assistance from a native English speaker with appropriate research background.

Point 3: The introduction should be rewritten to show the highlights and novelty of the work. Also, authors can cite the following work in the introduction which is related to their work and recently reported: "Simulation of melting and solidification of graphene nanoparticles-PCM inside a dual tube heat exchanger with extended surface." Journal of Energy Storage 44 (2021): 103265. "Numerical study of the effect of graphene nanoparticles in calcium chloride hexahydrate-based phase change material on melting and freezing time in a circular cavity with a triangular obstacle." Journal of Energy Storage 43 (2021): 103243.

Response 3: Thank you for underlining this deficiency. We have revised the introduction to reflect the highlights and significance of the review and quote the above two novel references nad have added to bibliography as references 3 and 4. (Line 43, 52; page 1; Line 635-637,page 21; Line 638,639, page 22)

Point 4: section of drawbacks and future could be increased quality of the manuscript.

Response 4: Thank you for your precious comments and advice. We have made corresponding changes in the introduction (Line 85, page 3) and the conclusion and outlook of the manuscript (Line 590, page 21).

Point 5: Maybe at the beginning of the article, there should be a list of abbreviations?

Response 5: Thank you for your comments, due to the limitation of space, our abbreviations have been annotated at the bottom of the pictures and tables in the article, which is also convenient for readers to find.

Point 6: A review paper not only should summarize recently published works, but also should contain critical and comprehensive discussions. Therefore, check the writing for the whole manuscript. The review should not be presented by listing what has been done by others.

Response 6: Thank you for your precious comments and advice. In each section or at the end of the article, we have made a brief analysis and summary, and also pointed out some existing difficulties and solutions. Such as : (Line 68, page1), (Line 103 ,130; page 4), (Line 178, page6), (Line 196, page7) , (Line 588, page20) etc. Which has been marked in red font.

Reviewer 3 Report

The authors reviewed the previous works regarding the preparation, structure, performances and applications of graphene based aerogels. Although the work is nicely presented, several improvements are required before it can be accepted for publications. 

1- The abstract is so messy, the authors didnt highlight the overall scope of the manuscript in the abstract. GCl ?? what is that mean? the abstract should be revised to contain more information to cover the overall scope of the work.

2- The keywords should be alphabetically organized. 

3- The quality and resolution of figures is too low, most of the figures are adapted from previous works with combining them or any further modifications. Review paper should contain at least 3 or 4 own drawing figures. 

4- The introduction should be one paragraph, introducing the background of the topic and present the research gab, why the authors want to publish this manuscript. 

5- Section 1.2, should be number 2,, it is too short, the authors should add one table to present different fabrication methods to compare between them. The authors mentioned the preparation in their title, they should review in details about the preparation approaches. 

6- Section 1.3 should be combined with the introduction (the last part), to show the novelty and scientific interest about graphene aerogels. 

7-  In the subsections of number 2, the discussion is too limited, the authors only present the results of previous works without any critical discussion or highlighting the prospective of the materials. 

8- Big number of figures is good in the comprehensive review, this manuscript lack the critical discussion and very limited number of works are presented here, the authors only present figure in each subsection. 

9- The application is very limited, the authors should include all the potential applications of the graphene based aerogels, since they are not presenting pure graphene aerogel. Numerous applications for graphene composites are available and should be included.

Author Response

Response to Reviewer 3 Comments

Point 1: The abstract is so messy, the authors didnt highlight the overall scope of the manuscript in the abstract. GCl ?? what is that mean? the abstract should brevised to contain more information to cover the overall scope of the work.

Response 1: We are extremely grateful to reviewer for pointing out this problem. We havecarefully revised the abstract and used GCA to represent graphene-based composite aerogel to unify the full text.

Point 2: The keywords should be alphabetically organized.

Response 2: Thank you for your suggestion. As suggested by reviewer we have finished sorting keywords in alphabetical order (Line 23, page 1).

Point 3: The quality and resolution of figures is too low, most of the figures are adapted from previous works with combining them or any further modifications. Review paper should contain at least 3 or 4 own drawing figures.

Response 3: Thank you for your precious comments and advice. We have improved the quality of the pictures, and re-merged and modified some pictures(Figure 5, Figure 8, Figure 12, Figure 13, Figure 16, Figure 19, etc. )

Point 4: The introduction should be one paragraph, introducing the background of the topic and present the research gab, why the authors want to publish this manuscript.

Response 4: Thank you for underlining this deficiency. This section was revised and modified according to the information showed in the work suggested by the reviewer (Line 85, page 3).

Point 5: Section 1.2, should be number 2,, it is too short, the authors should add one table to present different fabrication methods to compare between them. The authors mentioned the preparation in their title, they should review in details about the preparation approaches.

Response 5: Thank you for underlining this deficiency. We have created a new table in the description of the preparation method, and analyzed the advantages and disadvantages of the commonly used methods and a wide range of choices(Line 89, page 3). Further classified and summarized references. Due to the requirements of the length of the article, we also briefly analyzed and summarized the other tables in this article.

Point 6: Section 1.3 should be combined with the introduction (the last part), to show the novelty and scientific interest about graphene aerogels.

Response 6: Thank you for your precious comments and advice. We have revised the introduction(Line 48, page 2) and section 1.3(Line 103, page 4), and further summarized the research value and research status of graphene aerogel in a wide range of fields.

 Point 7: In the subsections of number 2, the discussion is too limited, the authors only present the results of previous works without any critical discussion or highlighting the prospective of the materials.

Response 7: Thank you for underlining this deficiency. This section was revised and modified according to the information showed in the work suggested by the reviewer (Line 85, page 3).

Point 8: Big number of figures is good in the comprehensive review, this manuscript lack the critical discussion and very limited number of works are presented here, the authors only present figure in each subsection.

Response 8: We are extremely grateful to reviewer for pointing out this problem. In each section or at the end of the article, we have made a brief analysis and summary, and also pointed out some existing difficulties and solutions. Such as : (Line 68, page1), (Line 103 ,130; page4), (Line 178, page6), (Line 196, page7) , (Line 519, page19) (Line 620, page21) (Line 588, page20) etc. Which has been marked in red font.

Point 9: The application is very limited, the authors should include all the potential applications of the graphene based aerogels, since they are not presenting pure graphene aerogel. Numerous applications for graphene composites are available and should be included.

Response 9: Our deepest gratitude goes to you for your careful work and thoughtful suggestions. Due to the limitation of the length of the article, our review of the application has not been fully carried out, but the summary is also very comprehensive, for example, in the aspect of adsorption, we not only introduce the selection of materials, adsorbent mechanism, including oil-water separation and so on. In the field of energy storage and sensing, we introduce its applications in supercapacitors, lithium batteries, lithium-sulfur battery electrode materials, stress sensors, flexible sensors and so on. Its application as heat insulation and flame retardant material is also introduced. In the field of biomedicine, we also introduce its potential applications in drug carriers, antimicrobials, biological scaffolds and biosensors.

Round 2

Reviewer 2 Report

The authors considered the comments of the reviewer. The revised manuscript is improved. However, because some parts are still obscure, the manuscript content is still not a quality fit for publication. I, therefore, recommend a new revision of the manuscript to bring its content up to a suitable level. 

I really would like to encourage the Authors to implement much more details to your article for better understanding not only by road engineers. Materials are not the dedicated journal for the road engineering society only. The potential readers might not achieve a good understanding of the information provided in the article if it is not well explained. My remarks are meant to help you improve the presentation of your valuable research.

The introduction is still not relevant and requires a comprehensive literature review.

Author Response

Response to Reviewer 2 Comments

Comments and Suggestions for Authors

   The authors considered the comments of the reviewer. The revised manuscript is improved. However, because some parts are still obscure, the manuscript content is still not a quality fit for publication. I, therefore, recommend a new revision of the manuscript to bring its content up to a suitable level. 

I really would like to encourage the Authors to implement much more details to your article for better understanding not only by road engineers. Materials are not the dedicated journal for the road engineering society only. The potential readers might not achieve a good understanding of the information provided in the article if it is not well explained. My remarks are meant to help you improve the presentation of your valuable research.

The introduction is still not relevant and requires a comprehensive literature review.

Respond:

Thank you very much for your comments.We have made major modifications to the introduction. Please see the red font in the introduction.

Merry Christmas to you.   

  1. Introduction

1.1. Graphene-based composite aerogel

Aerogel, a highly porous material with low density and high specific surface area, is obtained by replacing the liquid in wet gel with gas without significantly changing the structure and volume of the gel network. Graphene-based composite aerogel (GCA) is composed of graphene and its derivatives graphene oxide (GO) and reduced GO (rGO) with other matrix materials. Its functions are mainly derived from graphene and its derivatives (graphene-based materials), while its structure and volume stability are mainly determined by other matrix materials[1]. The research results indicate that GCA has lower density, higher porosity, smaller pore diameter, larger specific surface area, and more stable morphology compared to general aerogels, but more important it has some unique characteristics, such as higher heat resistance, better electrical conductivity, and higher absorbability of metal ions [2,3]. Therefore, GCA can be used not only as a thermal insulation, sound insulation, damping, and adsorptive material, but also as an electrode material for sensors and energy storage devices [4], which has become a research hotspot and attracted people’s attention in recent years. Figure 1 shows the structure, properties, and application of GCA, and Figure 2 displays a structural schematic of graphene-based materials [5].

1.2. Preparation principle of GCA

The preparation principle of GCA mainly includes three key processes: sol, gel, and drying. In the sol-gel process, the reactants are uniformly mixed and reacted in the liquid phase to form clusters of different structures. The sol of graphene-based materials and matrix may be obtained by chemical vapor deposition (CVD), hydrothermal reaction, chemical reduction, and vacuum carbonization [7]. Then composite gel is constructed by the self-assembly, chemical cross-linking, template method and 3D printing. The gel contains a large amount of water or other solvents, even more than 90%, with stable volume and no fluidity [8]. Then the solvent is removed from the gel by freeze-drying or supercritical drying to obtain GCA. The GCA always maintains higher porosity and larger specific surface area and has a similar network structure consisting of graphene-based and other matrix materials. Table 1 displays the preparation principle of GCA

1.3. Current research situation

Figure 3 shows the number of papers published on GCA from 2010 to 2021, indicating that the research began in 2010 and the number of papers grew exponentially in the last 10 years. According to the literature reports, GCA is usually composed of graphene and its derivatives and matrix materials. The properties and functions of GCA are mainly determined by graphene and its derivatives, whilethe porous structure and stability are mainly determined by matrix materials. The matrix materials include inorganic nanomaterials, synthetic polymers and natural polymers.  Therefore, the research process on the fabrication method, material selection, structure construction, performance and application design of GCA is very complex, and it is necessary to summarize and guide based on these research results. Nowsday there is also a lack of the targeted summary articles. In this review, we focus primarily on reviewing the fabrication of GCA with inorganic nanomaterials, synthetic polymer, and natural polymers along with its structure, performance, and applications.

Reviewer 3 Report

Although, the authors have improved the revised manuscript, few more things need to be done; 

1- "Figure 3. Number of papers on GCA in the last 12 years (as of 20 September 2021)" the authors should specify how they obtained these number, the type of database such as science direct, scopus, etc. 

2- Omit "in the literature" from table 1 capture. 

3- The authors added the section "1.2. Preparation methods of GCA", while there is a separate section regarding "2. Composition and preparation of GCA", this can be confusing to the readers, I suggest changing the titles of these sections depending on the content. Section 1.3 is general preparation approach, thus, putting a table for specific composite aerogel is not advisable, it should be more on the principal rather than the types. 

4- I found one recent review the author should cite in section 5.1, as one of the aerogel application in term of Adsorption removal of contaminants from water; https://doi.org/10.1016/j.jwpe.2021.102481

5- The cytotoxicity of Graphene based aerogel limitate its biomedical applications, the authors should address one paragraph about it and cite some articles about the cytotoxicity of graphene based materials. 

Author Response

Response to Reviewer 3 Comments

Point 1- "Figure 3. Number of papers on GCA in the last 12 years (as of 20 September 2021)" the authors should specify how they obtained these number, the type of database such as science direct, scopus, etc. 

Respond:

     We have added the database used for literature reference in Figure 3.  

Figure 3. Number of papers on GCA in the last 12 years (as of 20 September 2021 by Science Direct record).

Point 2- Omit "in the literature" from table 1 capture. 

Respond:

   The “in the literature” has be deleted from Table 1 capture.

Point 3- The authors added the section "1.2. Preparation methods of GCA", while there is a separate section regarding "2. Composition and preparation of GCA", this can be confusing to the readers, I suggest changing the titles of these sections depending on the content. Section 1.3 is general preparation approach, thus, putting a table for specific composite aerogel is not advisable, it should be more on the principal rather than the types. 

Respond:

We have modified "2.Composition and preparation of GCA" to "2.Composition of GCA", and hope to make some improvement.

Point 4- I found one recent review the author should cite in section 5.1, as one of the aerogel application in term of Adsorption removal of contaminants from water; https://doi.org/10.1016/j.jwpe.2021.10248

Respond:

We have added the paper recommended by reviewers as reference [75] to the corresponding section 5.1.  

[75] Mariana Mariana; Abdul KhalilH.P.S.; Esam BashirYahya;N.G.Olaiya;Tata Alfatahc;Suriani;Azmi Mohame.Recent trends and future prospects of nanostructured aerogels in water treatment applications.Journal of Water Process Engineering,2022,45,102481. https://doi.org/10.1016/j.jwpe.2021.10248

Point 5- The cytotoxicity of Graphene based aerogel limitate its biomedical applications, the authors should address one paragraph about it and cite some articles about the cytotoxicity of graphene based materials. 

     Respond:

we have introduced the study on the cytotoxicity of Graphene based aerogel and added references accordingly. Meanwhile, references on The cytotoxicity of Graphene based aerogel [111] [112] and [115] are added.  (section 5.4.).
